# JAK inhibitors dampen activation of interferon-stimulated transcription of ACE2 isoforms in human airway epithelial cells

Hye Kyung Lee [1✉], Olive Jung [2,3] & Lothar Hennighausen [1✉]

SARS-CoV-2 infection of human airway epithelium activates genetic programs leading to progressive hyperinflammation in COVID-19 patients. Here, we report on transcriptomes activated in primary airway cells by interferons and their suppression by Janus kinase (JAK) inhibitors. Deciphering the regulation of the angiotensin-converting enzyme 2 (ACE2), the receptor for SARS-CoV-2, is paramount for understanding the cell tropism of SARS-CoV-2 infection. ChIP-seq for activating histone marks and Pol II loading identified candidate enhancer elements controlling the *ACE2* locus, including the intronic *dACE2* promoter. Employing RNA-seq, we demonstrate that interferons activate expression of *dACE2* and, to a lesser extent, the genuine *ACE2* gene. Interferon-induced gene expression was mitigated by the JAK inhibitors baricitinib and ruxolitinib, used therapeutically in COVID-19 patients. Through integrating RNA-seq and ChIP-seq data we provide an in-depth understanding of genetic programs activated by interferons, and our study highlights JAK inhibitors as suitable tools to suppress these in bronchial cells.

[1] Laboratory of Genetics and Physiology, National Institute of Diabetes, Digestive and Kidney Diseases, National Institutes of Health, Bethesda, MD, USA. [2] Division of Preclinical Innovation, National Center for Advancing Translational Sciences, National Institutes of Health, Rockville, MD, USA. [3] Biomedical Ultrasonics & Biotherapy Laboratory, Institute of Biomedical Engineering, Department of Engineering Science, Old Road Campus Research Building, University of Oxford, Headington, Oxford, UK. ✉email: hyekyung.lee@nih.gov; lotharh@niddk.nih.gov

The angiotensin-converting enzyme 2 (ACE2) receptor is the gateway for SARS-CoV-2 to airway epithelium[1,2] and the strong inflammatory response after viral infection is a hallmark in COVID-19 patients. Single cell RNA sequencing (scRNA-seq) studies have revealed that *ACE2* expression in type II pneumocytes is induced by interferons[3–7], suggesting that the presence of an autoregulatory loop could result in increased viral

infection. Recent studies[8–10] have identified a hitherto unknown short form of ACE2, called dACE2, that originates from an intronic promoter activated by interferons, calling into question the earlier studies that transcription of the full length native *ACE2* is under interferon control. Onabajo and colleagues[9] used ENCODE data for histone modification marks (H3K4me3, H3K4me1, and H3K27ac) as well as DNase I hypersensitive

**Fig. 1 Expression of *ACE2* isoforms is induced by interferons. a–c** ACE2, and STAT1 and ISGs mRNA levels from control cells and cells treated with different cytokines were measured by RNA-seq. Results are shown as the means ± s.e.m. of independent biological replicates ($n = 3$). One-way ANOVA followed by Dunnett's multiple comparisons test was used to evaluate the statistical significance of differences. **d** RNA-seq reads were matched with each exon, including the new exon (ex1c), of the *ACE2* gene. The *CELF1* gene served as a control for the quality of the RNA-seq data. **e** SAECs were cultured in the absence and presence of interferon alpha (IFNα) and beta (IFNβ) followed by RNA-seq analyses. The reads covering key exons (1a, 1b, 9, 1c and 10) are shown. **f** mRNA levels of exon9 and exon1c were measured using qRT-PCR and normalized to GAPDH levels. Results are shown as the means ± s.e.m. of independent biological replicates (Control and IFNβ, $n = 9$; IFNα, $n = 3$). Two-way ANOVA with followed by Tukey's multiple comparisons test was used to evaluate the statistical significance of differences between control and cytokine-treated cells.

(DHS) sites in cell lines to mark putative regulatory elements at the newly identified exon (ex1c) located within intron 9 of the *ACE2* gene. However, no regulatory elements were detected in the vicinity of the 5′ end of the full-length transcript encoding biologically active ACE2, and in sequences distal to the genuine promoter. Since these data sets were obtained from a wide range of cell lines and not from human primary airway cells, the principal target of SARS-CoV-2, they might not present a comprehensive picture of the regulatory regions controlling expression of the entire *ACE2* locus, including *dACE2*, in bronchial tissue.

Janus kinase (JAK) inhibitors suppress signal transduction pathways leading to immune activation and inflammation and they are used therapeutically to harness cytokine storms in COVID-19 patients[11,12]. However, the genetic programs activated in airway cells by interferons and their response to JAK inhibitors remain to be elucidated. To better understand these programs, we investigated the transcriptomes activated by interferon α, β, γ, and λ as well as the cytokines growth hormone (GH), IL6, and IL7 in human primary airway cells. Using ChIP-seq for active histone marks and Pol II loading, we identified candidate enhancer elements controlling genes regulated by the different interferons. This also led to a better understanding of the structure and regulation of the *ACE2* locus, including the intronic promoter encoding the *dACE2* transcript. Our study also highlights that JAK inhibitors are suitable tools that efficiently blunt interferon-activated genetic programs.

## Results

**Interferons regulate the *ACE2* locus in human airways cells.** To comprehensively identify the genetic elements controlling the extended *ACE2* locus, with an emphasis on its response to interferons, we focused on human primary Small Airway Epithelial Cells (SAECs) expressing a wide range of cytokine receptors and key mechanistic components of the executing JAK/STAT signal transduction pathway (Supplementary Data 1). We stimulated SAECs with interferon type I (IFNα and IFNβ), type II (IFNγ), and type III (IFNλ) as well as with growth hormone (GH), Interleukin 6 (IL6) and IL7, followed by RNA-seq transcriptome analyses (Supplementary Data 2–8). *ACE2* expression increased ~16 to 18-fold upon stimulation with the four interferons but not with GH, IL6 and IL7 (Fig. 1a). This degree of induction was similar to some classical interferon stimulated genes (ISG), such as *STAT1* and *ISG20* (Fig. 1b–c). In contrast, expression of *ISG15* was induced by approximately 500-fold (Fig. 1c).

Our RNA-seq data sets (Supplementary Data 2–8) created a foundation to dig deeper and comprehensively investigate gene classes differentially activated by cytokines in SAECs. IFNα treatment resulted in a more than 2-fold activation of 1405 genes, 647 genes were induced by IFNβ, 1570 by IFNγ, and 1285 by IFNλ3. IL6 activated 1432 genes, IL7 1782 genes, and GH 2807 genes. A total of 229 genes (Supplementary Data 9) were activated by all four interferons, and four genes (IRF7, IFI27, IFI35, and

PARP10) were induced by all seven cytokines used in this study. Gene Set Enrichment Analysis (GSEA) showed, depending on the interferon, that between 17% and 26% of the induced genes were linked to immune responses, TNFα signaling, inflammatory response, IL-JAK/STAT signaling, among others (Supplementary Fig. 1a and Supplementary Data 2–5). While the family of interferon induced genes (IFIs) was, in general, activated by all four interferons, the magnitude differed more than 10-fold between individual genes and gene families (Supplementary Fig. 2). Notably, genes encoding the family of Interferon-induced proteins with tetratricopeptide repeats (IFIT) were induced up to several thousand-fold. In contrast activation of the gene family of interferon-induced transmembrane proteins (IFITM) was at least one order of magnitude lower. Other studies used bulk RNA-seq of primary human alveolar epithelial type-2 cells (AT2s) infected with SARS-CoV-2[7] and scRNA-seq from human bronchial epithelium (BEAS-2B) stimulated with interferons[5]. There was an overlap between genes induced by interferons in SAECs and SARS-CoV-2 in AT2s (Supplementary Fig. 2). In contrast, scRNA-seq provides less read depth and a direct comparison with bulk RNA-seq is therefore not possible. As a positive control, we also measured the activation of *STAT1*, *STAT2*, *SOCS1 SOCS3*, and *SIRF1* and *IRF7* (Supplementary Fig. 3e–f). The cytokines IL6, IL7, and GH activated different gene sets (Supplementary Data 6–8). Our results not only identified gene sets activated by interferons but also demonstrated their differential response to individual interferons (Supplementary Fig. 3).

The *ACE2* locus encodes two mRNAs, one encoding the full-length ACE2 and one encoding the N-terminally truncated form of ACE2 (dACE2). The *dACE2* mRNA is initiated from an intronic promoter and in agreement with earlier studies[8,9], we detected the *dACE2* exon (ex1c) within intron 9 of the *ACE2* gene (Fig. 1d). To obtain additional information on the interferon response of the *dACE2* and *ACE2* promoters, we used RNA-seq and determined the respective read counts over the three alternative first exons (Fig. 1d–e). While the increase of RNA-seq reads induced by IFNα/β was highest (~25-fold) over ex1c, a significant, ~2–10-fold increase was detected over ex1a and ex1b, supporting the notion that expression of the full-length *ACE2* transcript is also under interferon control. The *CELF1* gene served as a control (Fig. 1d). As an independent assay we used qRT-PCR and determined that IFN α/β stimulation led to an 8 to 15-fold increase of *dACE2* (ex1c) and an approximately ~3-fold increase of *ACE2* RNA (ex9) (Fig. 1f). In addition, both AEC2 isoforms were detected by Western blot (Supplementary Fig. 4). Previous studies in normal human bronchial epithelium (NHBE) did not reveal an interferon response of the native *ACE2* promoter[8,9] suggesting differences between cell types or culture conditions. The mouse *Ace2* gene is also induced by cytokines through a JAK/STAT5-activated enhancer in the second intron[13] and a DHS site is located in the equivalent location on the human *ACE2* gene in SAECs and lung tissue. This suggests the presence of additional regulatory elements controlling expression of the full-length ACE2 mRNA.

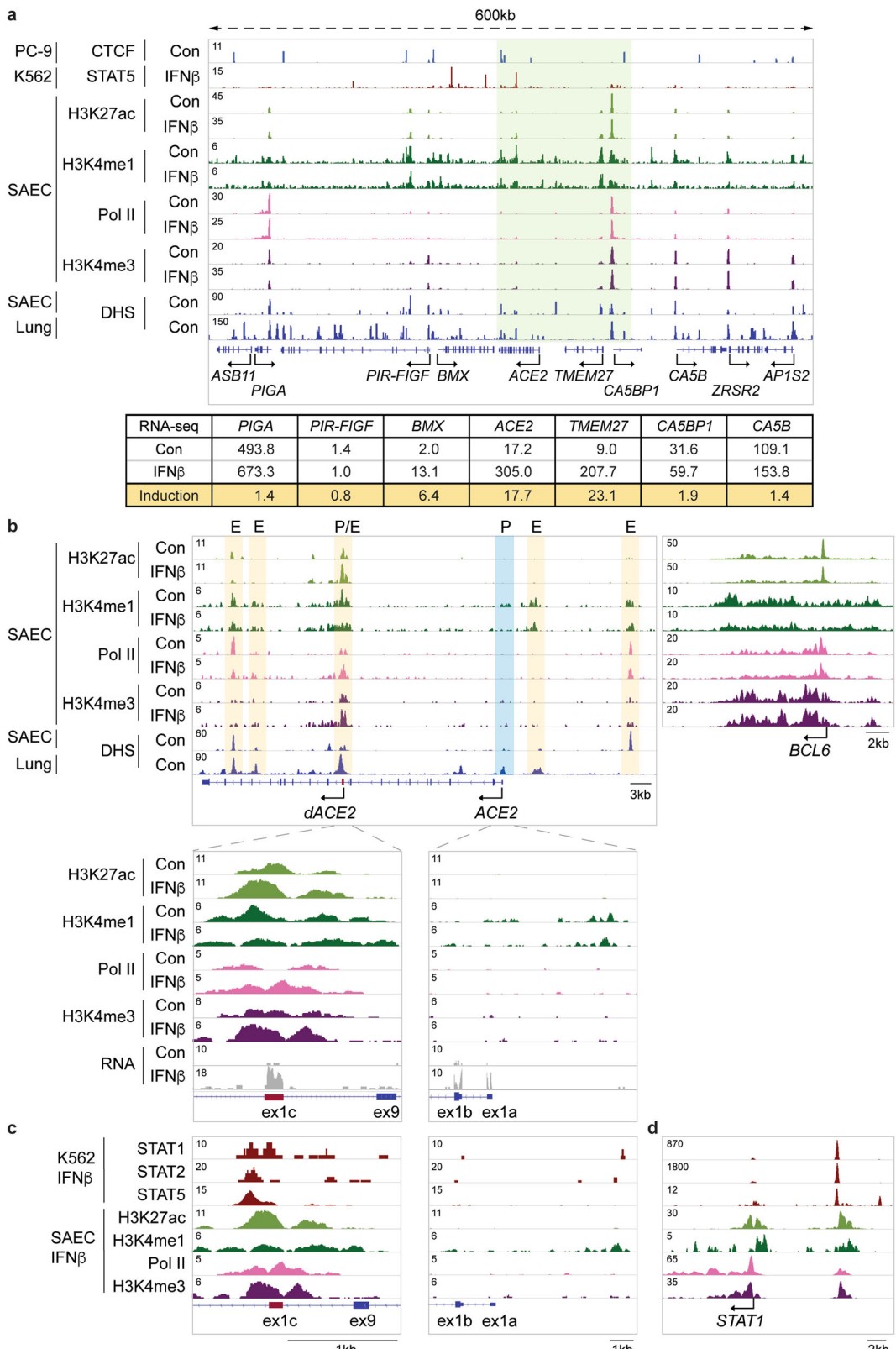

| RNA-seq | PIGA | PIR-FIGF | BMX | ACE2 | TMEM27 | CA5BP1 | CA5B |
|---|---|---|---|---|---|---|---|
| Con | 493.8 | 1.4 | 2.0 | 17.2 | 9.0 | 31.6 | 109.1 |
| IFNβ | 673.3 | 1.0 | 13.1 | 305.0 | 207.7 | 59.7 | 153.8 |
| Induction | 1.4 | 0.8 | 6.4 | 17.7 | 23.1 | 1.9 | 1.4 |

**Interferons activate enhancers in the *ACE2* locus**. To identify candidate regulatory elements controlling the extended *ACE2* locus, including *ACE2* and *dACE2*, in primary airway cells, we conducted ChIP-seq for the histone marks H3K27ac (activate loci), H3K4me1 (enhancers), and H3K4me3 (promoters). We also probed RNA Polymerase II (Pol II) loading in SAECs in the absence and presence of IFNβ (Fig. 2). DNase I hypersensitive (DHS) sites from human lung tissues[14] and SAECs[15] served as bona fide predictors of regulatory regions. In addition to *ACE2*, expression of the neighboring *TMEM27* gene, an *ACE2* homologue, as well as *BMX* were activated by interferons (Fig. 2a) suggesting the presence of shared regulatory elements. *ACE2* and *TMEM27* originated from a gene duplication and their response to interferon is equivalent. The positions of the chromatin

Fig. 2 Structure of the extended *ACE2* locus and regulation of the *ACE2* locus in primary airway epithelial cells. a Regulatory marks in the 600 kb locus including *ACE2* and neighboring genes in SAECs. mRNA levels in the absence and presence of IFNβ were measured by RNA-seq. ChIP-seq for the histone marks H3K4me1, H3K27ac, and H3K4me3, and Pol II was conducted in SAECs in the absence and presence of IFNβ. CTCF and STAT5 ChIP-seq data and DHS data are from ENCODE. b ChIP-seq experiments for the histone marks, H3K4me3 (promoter), H3K4me1 (enhancers), H3K27ac (active genes), and Pol II loading. The DHS data were obtained from ENCODE[14,15]. Yellow shade, candidate enhancers and blue shade, predicted promoter. The P/E region within intron 9 probably constitutes a combined promoter/enhancer unit. The *BCL6* gene severed as ChIP-seq control. Solid arrows indicate the orientation of genes. c A putative STAT5 enhancer in the *ACE2* gene was identified using ChIP-seq data from IFNβ-treated K562 cells[16]. d ChIP-seq data for STAT transcription factors, histone marks H3K4me3 (promoter), H3K4me1 (enhancers) and H3K27ac (active genes), and RNA Pol II at the *STAT1* locus.

boundary factor CTCF suggests that *ACE2* and *TMEM27* are located within a sub-TAD (Fig. 2a).

In agreement with earlier studies[9], we identified DHS sites at ex1c located in intron 9 and in intron 17 (Fig. 2b). In addition, we identified DHS sites in the vicinity of ex1a, likely marking the genuine *ACE2* promoter, a distal site marking a possible enhancer and one in intron 15 coinciding with activate chromatin marks. The DHS site in intron 9 overlaps with strong H3K4me3 marks, identifying it as a genuine promoter region (Fig. 2b). This site is also decorated with H3K4me1 and H3K27ac marks and extensive Pol II loading, hallmarks of a complex promoter/enhancer. H3K27ac and Pol II loading was further induced by IFNβ, reflecting increased *ACE2* expression. The cytokine regulated *BCL6* gene served as a control. IFNβ activated the transcription factors STATs 1, 2, and 5 (Fig. 1b and Supplementary Data 3) and ChIP-seq experiments from K562 erythroid cells stimulated with IFNβ[16] revealed preferential binding of STAT5 to the intronic promoter/enhancer (Fig. 2c) further supporting regulation through the JAK/STAT pathway. The *STAT1* locus served as a control for the binding of STAT transcription factors (Fig. 2d). While the *ACE2* promoter associated with ex1a is marked by a DHS site and H3K4me1 marks, there is little evidence of H3K4me3 and H3K27ac marks. However, it is well known that there is no direct relationship between gene activity and the presence of these marks.

Based on the presence of DHS sites, activating histone marks and Pol II loading, either in combination or by themselves, several candidate enhancers regulating the *ACE2* locus were identified (Fig. 2b). In addition, activating histone marks and DHS sites[14,15] marked candidate regulatory elements in other loci under interferon control, such as *STAT1* and *ISG15*, in human lung tissues (Fig. 3)[5,17]. Members of the IFIT gene family were induced several thousand-fold by interferons (Supplementary Fig. 3) and ChIP-seq profiled revealed the regulatory landscape in this genetic locus (Supplementary Fig. 5). Candidate regulatory elements were also identified in *IRF1* and 9, *STAT1*, *IFI44* and members of the *CXCL* gene family. Our RNA-seq and ChIP-seq data should foster research aimed at understanding interferon-regulated genetic programs in airway cells.

**JAK inhibitors suppress the IFNβ-sensing enhancers in the *ACE2* locus.** Interferons activate genetic programs through the JAK/STAT signaling pathway and JAK inhibitors are used clinically in COVID-19 patients in an effort to suppress the genomic consequences of cytokine storms[11,18,19]. To investigate if interferon-induced *ACE2* and *dACE2* expression is controlled by the JAK/STAT pathway, we cultured SAECs in the presence of IFNβ and the JAK inhibitors, baricitinib and ruxolitinib, followed by RNA-seq, qRT-PCR assays (Fig. 4a–c and Supplementary Data 10–11) and ChIP-seq analyses (Fig. 4e–h). Both JAK inhibitors blunted the IFNβ-induced increase of the full-length *ACE2* (ex1a, ex1b, and ex9) and the *dACE2* (ex1c) transcripts (Fig. 4a–b), supporting that their respective promoters are under JAK/STAT control. The efficacy of the two inhibitors extended to

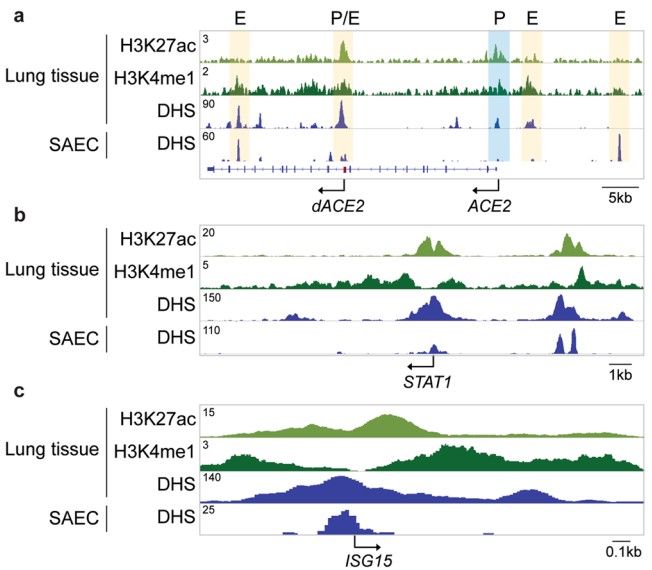

**Fig. 3 Activating histone marks at the *ACE2* locus in lung tissue.** ChIP-seq for histone marks H3K4me1 (enhancers) and H3K27ac (active genes), and DHS from human lung tissues and DHS from SAECs display regulatory elements at the *ACE2* (**a**), *STAT1* (**b**), and *ISG15* (**c**) loci.

a range of genetic programs activated through the pan JAK/STAT pathway (Supplementary Fig. 1b) and induction of bona fide interferon stimulated genes (ISG), such as *STAT1* and *ISG15*, was suppressed (Fig. 4c, d). In *ACE2*, as in other ISGs, ruxolitinib treatment mitigated the establishment of activating H3K27ac marks and Pol II loading over distal and intronic regulatory elements (Fig. 4e–h). IFNβ-induced expression of *ISG*, *IFIT*, and *IFITM* (Supplementary Fig. 3) was suppressed by baricitinib and ruxolitinib (Supplementary Data 9; Fig. 5). In addition to key transcription factors, primary RNA pattern recognition receptors (PRRs), including (RIG-I)-like receptors, are critical for the interferon production by lung epithelial cells upon infection by SARS-CoV-2[20]. Specifically, LGP2[21] (DHX58) and MDA5 are receptors binding double-stranded viral RNA intermediates and they primarily regulate IFN induction in response to SARS-CoV-2[20]. Notably, *LGP2* itself was induced by interferons and this activation was mitigated by JAK inhibitors (Supplementary Figs. 3h and 5).

While elevated interferon levels following the initial virus infection modulate the host's immune system, persistent upregulation can lead to cytokine storms and more severe symptoms[22,23]. To address the efficacy of JAK inhibitors in the suppression of *ACE2* expression, we performed additional kinetic experiments. The expression of *ACE2* and *STAT1* significantly increased within four hours after IFNβ stimulation and peaked at 12 h (Fig. 6a). Increased expression from the *dACE2* promoter accounted for most of elevated *ACE2* levels. This IFNβ-induced gene expression was suppressed by the JAK inhibitor ruxolitinib and persisted for 12 h (Fig. 6b). Lastly, pretreatment of cells with

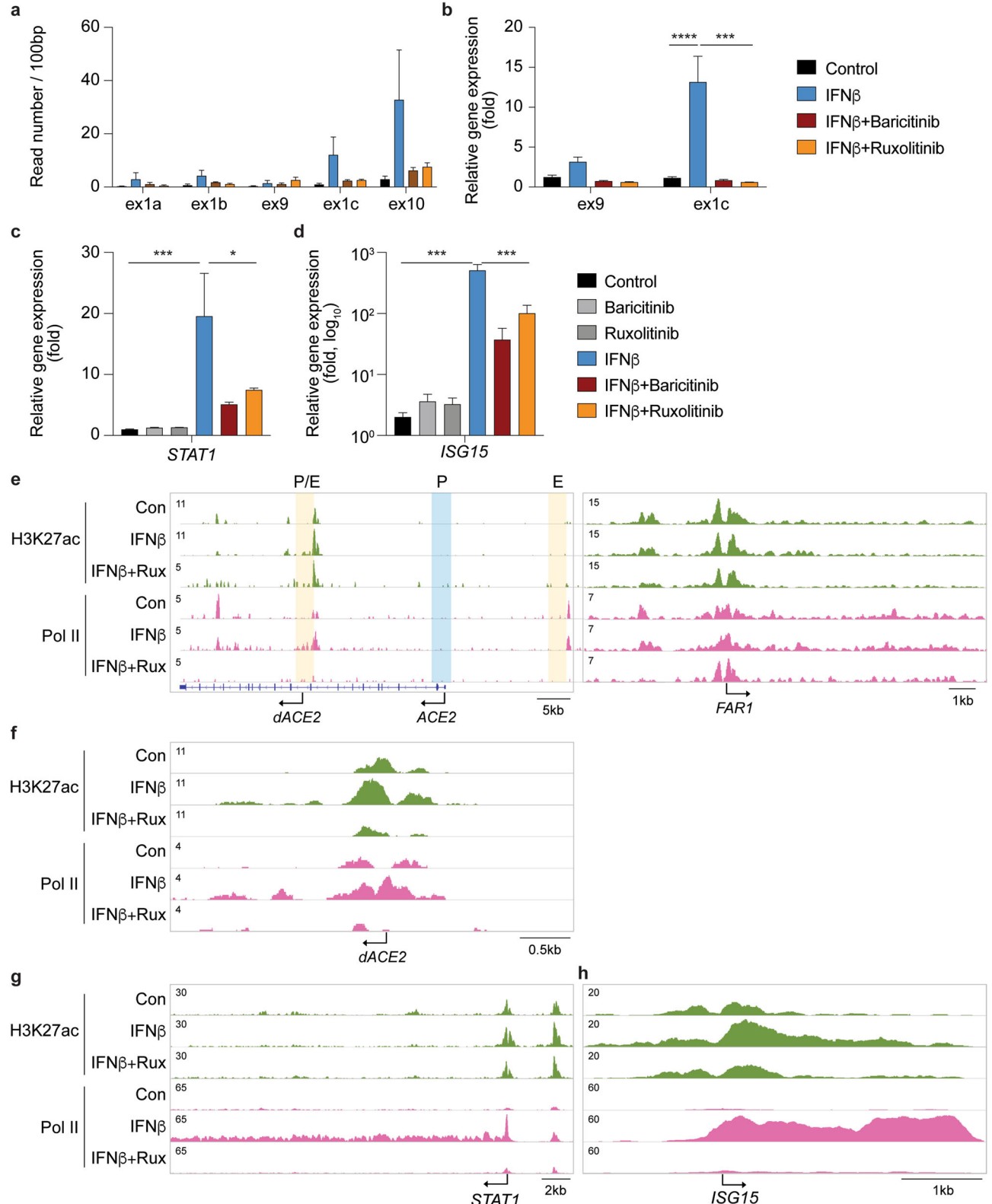

**Fig. 4 JAK inhibitors mitigate activation of IFNβ-stimulated genes. a–b** SAECs were cultured in the presence of IFNβ and JAK inhibitors, either baricitinib or ruxolitinib followed by RNA-seq analyses (**a**) and qRT-PCR (**b**). Reads covering key exons are displayed. qRT-PCR results were normalized to GAPDH levels and are shown as the means ± s.e.m. of independent biological replicates (Control and IFNβ, $n = 9$; IFNβ+JAK inhibitors, $n = 3$). Two-way ANOVA with followed by Tukey's multiple comparisons test was used to evaluate the statistical significance of differences between control and cytokine/Jak inhibitor-treated cells. **c–d** STAT1 and ISG15 mRNA levels from control and experimental cells were measured by RNA-seq. Results are shown as the means ± s.e.m. of independent biological replicates ($n = 3$). One-way ANOVA with followed by Dunnett's multiple comparisons test was used to evaluate the statistical significance of differences between control and cytokine/JAK inhibitor-treated cells. **e–h** H3K27ac marks, and Pol II loading at the *ACE2*, *STAT1*, and *ISG15* loci in SAECs in the absence and presence of IFNβ and the JAK inhibitor, ruxolitinib. The *FAR1* locus was used as a ChIP-seq control.

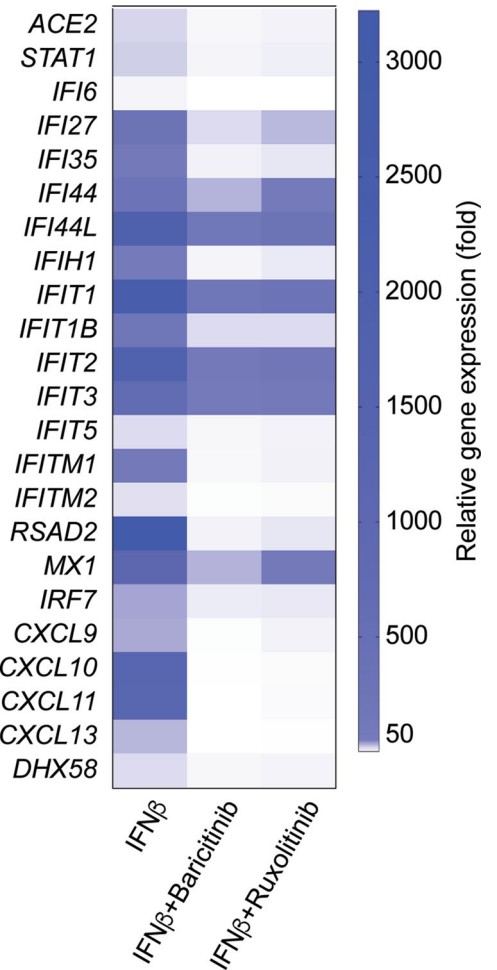

**Fig. 5 JAK inhibitors suppress distinct interferon-regulated immune pathways.** Heat map comparison of relative expression levels of representative interferon-responsive genes in SAECs stimulated with IFNβ (n = 3) alone or with IFNβ and JAK inhibitors (n = 3).

ruxolitinib for two hours abrogated the interferon response of ACE2 and STAT1 (Fig. 6c). JAK inhibitors efficiently suppress IFN-induced gene expression; these results mechanistically clarify that JAK inhibitors are therapeutically beneficial in COVID-19 patients, including those experiencing cytokine storms.

## Discussion

SARS-CoV-2 entry into nasal epithelia and lung tissue is facilitated by ACE2 and deciphering the regulation of the ACE2 gene will aid our understanding of the viral tropism and also non-pulmonary pathology. Here, we have identified candidate regulatory elements in the extended ACE2 locus and propose roles in controlling gene expression induced by interferons and other, not yet defined, stimuli. The intronic region associated with the dACE2 transcript[8–10] bears promoter and enhancer marks and binds STAT transcription factors, suggesting a complex regulatory element controlling interferon-induced expression. Additional candidate enhancers were identified distal to the genuine ACE2 promoter and within introns[3,4]. Our high-resolution bulk RNA-seq data sets on differentiated human primary small airway epithelial cells (SAECs) revealed the presence of the genuine full-length ACE2 transcript and the short form of ACE2, named dACE2, which had been identified in other studies using bulk RNA-seq[8–10]. Our transcriptome studies and qRT-

PCR demonstrate that interferons strongly activate expression of dACE2 but only marginally ACE2. dACE2 encodes a non-functional protein, which does not bind SARS-CoV-2[9,10], strongly suggesting that the interferon response does not result in increased viral infection. However, further studies are needed to investigate whether dACE2 could be a decoy and thus impede cellular uptake of SARS-CoV-2 upon interferon treatment. Single cell RNA-seq (scRNA-seq) studies[3–7] have shown elevated ACE2 expression in primary human airway cells stimulated with type I interferons. Based on these findings a feedback loop hypothesis was formulated suggesting that increased ACE2 levels induced by the cytokine storm would result in an increased infectivity. However, current evidence suggests interferons largely activate expression of the non-functional dACE2 isoform. In general, the lower sequencing depth of scRNA-seq makes it more challenging to identify alternative transcripts. Whether ACE2 RNA levels correlate with protein levels remains an open question[24]. Since previous scRNA-seq studies did not detect the more abundant dACE2 transcript[3–7,24], which would encode a protein lacking the 356 N-terminal amino acids, additional studies are needed.

Our ChIP-seq data also point to several additional candidate enhancers controlling the entire ACE2 locus, including the neighboring TMEM27 gene. Further studies are warranted to investigate ACE2 regulation in other SARS-CoV-2 target cells, such as pancreas[25,26], gut[27], and mammary epithelium. While the regulation of the human and mouse ACE2[13] loci displays distinct differences, they share their response to cytokines and the JAK/STAT pathway, with the mouse Ace2 gene being activated in mammary tissue by cytokines through an intronic enhancer[13]. Further studies in tissues and primary cells are needed to better understand the complex, possibly cell-specific, regulation of the human ACE2 locus in organs with extrapulmonary manifestations of SARS-CoV-2 infection.

Type I interferon (IFNα and IFNβ) secreted by SARS-CoV-2-infected airway epithelium controls autocrine and paracrine genetic programs. Here, we used small airway epithelial cells (SAECs) that, like human bronchial epithelial cells (HBEs), differentiate in air liquid interphase (ALI) culture systems and form tight junctions and displaying cilia. Other studies have focused on primary tonsil epithelial cells[9], primary normal human bronchial epithelial cells (NHBE)[8–10], nasal epithelial cells (NECs)[10], BEAS-2B cells (bronchial epithelium)[5], primary human air way epithelial cells from human donors[5] as well as HEK293T, A549, SCC-4, SCC-25, Vero, CV-1, MDCK, R9ab, and MCA-38 cell lines[8]. While any of these cells respond to cytokines, the magnitude of the transcriptome response likely differs between experimental systems and is dependent on the presence and abundance of various components of the JAK/STAT signaling pathway. Since some studies used bulk RNA-seq and others scRNA-seq a definitive comparison of the genetic programs induced by interferons is difficult to achieve. Bulk RNA-seq is an appropriate approach to gauge interferon-activated genetic programs as indicated by the up to 50,000-fold induction of the IFIT gene family encoding antiviral proteins. Integrating the comprehensive RNA-seq and ChIP-seq data sets also permits the genome-wide identification of candidate enhancers responsive to interferons and other cytokines.

JAK inhibitors[28–30] are used therapeutically to treat COVID-19 patients[11,12,31] and our studies have identified IFN-induced genetic pathways that are suppressed by ruxolitinib and baricitinib. Both JAK inhibitors had an equivalent capacity to effectively mitigate IFNβ-induced gene activation and blunt immune signaling pathways. Autosomal-recessive and autosomal-dominant mutations in interferon pathway genes have been identified in some patients with life-threatening COVID-19[32] and high TYK2 expression was linked to COVID-19 disease

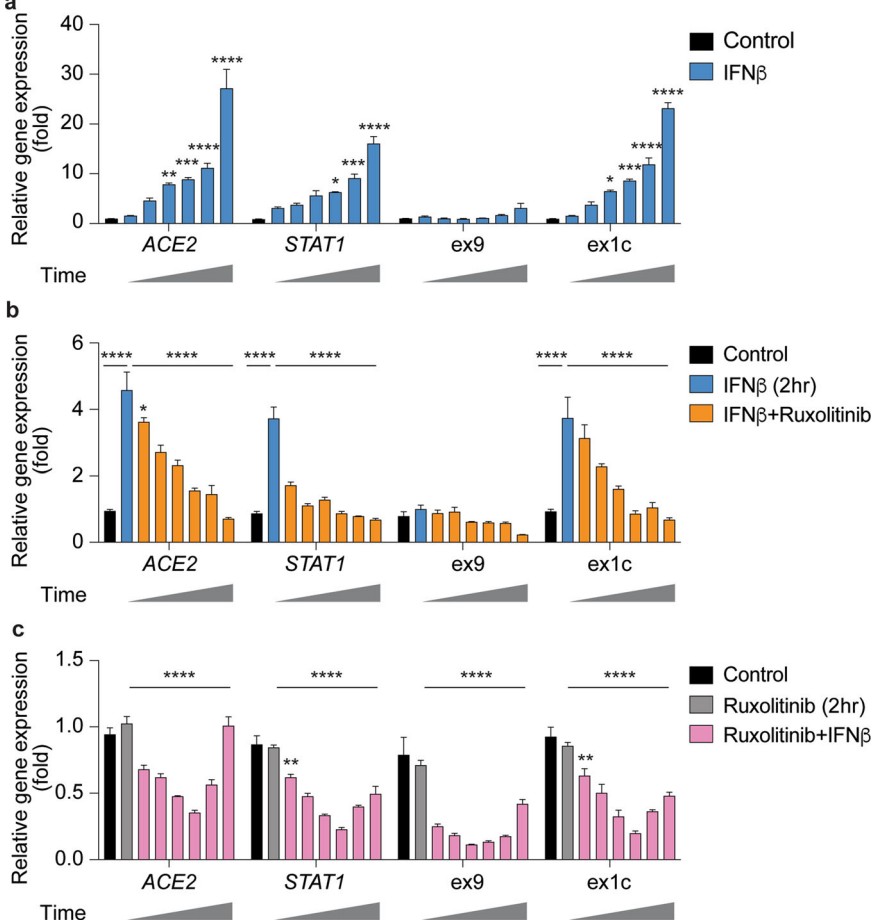

**Fig. 6 JAK inhibitor ruxolitinib suppresses IFNβ-stimulation. a** SAECs were stimulated with IFNβ for 12 h (0, 1, 2, 4, 6, and 12 h) and analyzed by qRT-PCR. **b** SAECs were cultured for two hours in the presence of IFNβ followed by the addition of ruxolitinib and qRT-PCR after 0–12 h (0, 1, 2, 4, 6, and 12 h). **c** SAECs were pretreated with ruxolitinib for two hours before stimulating the cells with IFNβ, followed by qRT-PCR for 0–12 h (0, 1, 2, 4, 6, and 12 h). qRT-PCR results were normalized to GAPDH levels and are shown as the means ± s.e.m. of independent biological replicates (all conditions, $n = 4$). Two-way ANOVA with followed by Tukey's multiple comparisons test was used to evaluate the statistical significance of differences between control and cytokine/Jak inhibitor-treated cells.

severity[33]. The promising clinical results with JAK inhibitors, which dampen the inflammation, and molecular studies, including ours, on the efficacy of JAK inhibitors in suppressing interferon-induced genetic programs has practical implications for their use in treating COVID-19 patients.

## Methods

**Cell culture**. Human small airway epithelial cells (SAECs) obtained from Lifeline Technology (FC-0016) were expanded using the complete BronchiaLife™ media kit (Lifeline Technology, LL-0023). All culture wares were pre-coated in 30 μg/ml of Fibronectin (ThermoFisher Scientific, 33016015) for at least 1 h at room temperature. Calu-3 line (ATCC, HTB-55™) were cultured using Eagle's Minimum Essential Medium (ATCC, 30-2003™) containing 10% fetal bovine serum (Cytiva, SH3007103) in 5% $CO_2$ atmosphere at 37 °C.

Cytokines (10 ng/ml; Human IFNβ, 300-02BC; Human IFNγ, 300-02; Human IL6, 200-06; Human IL7, 200-07; Human Growth hormone, 100-40, Peprotech; Human IFNα2b, 78077.1, Stem Cell Technologies; Human IFNλ3, 5259-IL-025, R&D systems) were added in respective culture media after serum starvation for 2 h, and the cells were incubated for 12 h in 5% $CO_2$ atmosphere at 37 °C. The cells were washed with PBS (Gibco, 14190144) twice and harvested.

Jak inhibitors, 10 μM of either Baricitinib (HY-15315A, MedChemExpress) or Ruxolitinib (HY-50856A, MedChemExpress), were added to BronchiLiafe™ media with or without IFNβ. SAECs and incubated for 12 h and then washed with PBS twice and harvested.

**RNA isolation and quantitative real-time PCR (qRT–PCR)**. Total RNA was extracted from the collected cells and purified using the PureLink RNA Mini Kit (Invitrogen) according to the manufacturer's instructions. cDNA was synthesized

from total RNA using Superscript II (Invitrogen). Quantitative real-time PCR (qRT-PCR) was performed using TaqMan probes (ACE2, Hs01085333_m1; STAT1, Hs01013996_m1; GAPDH, Hs02786624_g1, Thermo Fisher scientific) on the CFX384 Real-Time PCR Detection System (Bio-Rad) according to the manufacturer's instructions. Exon9 and exon1c mRNA were measured with the following primers, that were used by Onabajo et al.[9], using TaqMan expression assay: Forward, 5′-GGGCGACTTCAGGATCCTTAT-3′, Reverse, 5′-GGATATGCC CCATCTCATGATGG-3′, Probe, 5′-ATGGACGACTTCCTGACAG-3′; Forward, 5′-GGAAGCAGGCTGGGACAAA-3′, Reverse, 5′-AGCTGTCAGGAAGTCGTC CATTG-3′, Probe, 5′-AGGGAGGATCCTTATGTG-3′. PCR conditions were 95 °C for 30 s; 95 °C for 15 s, and 60 °C for 30 s for 40 cycles. All reactions were done in triplicate and normalized to the housekeeping gene GAPDH. Relative differences in PCR results were calculated using the comparative cycle threshold ($C_T$) method.

**Total RNA sequencing (Total RNA-seq) and data analysis**. Total RNA was extracted from the collected cells and purified using the PureLink RNA Mini Kit (Invitrogen) according to the manufacturer's instructions. Ribosomal RNA was removed from 1 μg of total RNAs and cDNA was synthesized using SuperScript III (Invitrogen). Libraries for sequencing were prepared according to the manufacturer's instructions with TruSeq Stranded Total RNA Library Prep Kit with Ribo-Zero Gold (Illumina, RS-122-2301) and paired-end sequencing was done with a HiSeq 3000 instrument (Illumina).

The raw data were subjected to QC analyses using the FastQC tool (version 0.11.9) (https://www.bioinformatics.babraham.ac.uk/projects/fastqc/). Total RNA-seq read quality control was done using Trimmomatic[34] (version 0.36) and STAR RNA-seq[35] (version STAR 2.5.4a) using 50 bp paired-end mode was used to align the reads (hg19). HTSeq[36] (version 0.9.1) was to retrieve the raw counts and subsequently, R (https://www.R-project.org/), Bioconductor[37], and DESeq2[38] were used. Additionally, the RUVSeq[39] package was applied to remove confounding factors. The data were pre-filtered keeping only those genes, which have at least ten

reads in total. The visualization was done using dplyr (https://CRAN.R-project.org/package=dplyr) and ggplot2[40]. Genes were categorized as significantly differentially expressed with an adjusted *p*-value (pAdj) below 0.05 and a fold change > 2 for upregulated genes and a fold change of < −2 for downregulated ones. Gene Set Enrichment Analysis (GSEA, https://www.gsea-msigdb.org/gsea/msigdb) were performed. Sequence read numbers were calculated using Samtools[41] software with sorted bam files.

**Chromatin immunoprecipitation sequencing (ChIP-seq) and data analysis.** Chromatin was fixed with formaldehyde (1% final concentration) for 15 min at room temperature, and then quenched with glycine (0.125 M final concentration). Chromatin washed with PBS was homogenized using a tissue grinder in Farnham's lysis buffer and fragmented by sonication. Fragmentation efficiency in the range of 250–500 bp was checked using gel electrophoresis and ChIP was performed with Dynabeads Protein A (Invitrogen) and the following antibodies (5–10 µg) were used for ChIP-seq: H3K27ac (Abcam, ab4729), RNA polymerase II (Abcam, ab5408), H3K4me1 (Active Motif, 39297), and H3K4me3 (Millipore, 07-473). Libraries for next-generation sequencing were prepared and sequenced with a HiSeq 3000 instrument (Illumina).

The raw data were subjected to QC analyses using the FastQC tool (version 0.11.9) (https://www.bioinformatics.babraham.ac.uk/projects/fastqc/). Quality filtering and alignment of the raw reads was done using Trimmomatic[34] (version 0.36), Bowtie[42] (version 1.2.2), and Samtools[41] (version 1.8), with the parameter '-m 1' to keep only uniquely mapped reads, using the reference genome hg19. Picard tools (Broad Institute. Picard, http://broadinstitute.github.io/picard/. 2016) was used to remove duplicates. Homer[43] (version 4.8.2) and DeepTools[44] (version 3.1.3) software was applied to generate bedGraph files, separately. Integrative Genomics Viewer[45] (version 2.5.3) was used for visualization. Each ChIP-seq experiment was conducted for more than two replicates. DeepTools was used to obtain the Pearson and Spearman correlation between the replicates.

**Western blot.** Proteins (100 µg) from SAECs extracted using lysis buffer (50 mM Tris-Cl pH 8.0, 150 mM NaCl, 0.5% Na-DOC, 1% NP-40, 0.1% SDS, 5 mM EDTA, 1 mM PMSF, and protease inhibitor cocktail) were separated on a 4–12% NuPage gradient gel (Invitrogen) and transferred to a PVDF membrane (Invitrogen). Membranes were blocked for 1 h with 5% nonfat dry milk in PBS-T buffer (PBS containing 0.1% Tween 20) and incubated for 1.5 h at 4 °C with the primary antibody (1:1000) against ACE2 (Proteintech, 21115-1-AP) and GAPDH (Cell signaling, #5174). After washing, membranes were incubated for 1 h with HRP-conjugated secondary antibodies (Cell signaling). Labeled protein bands were detected using an enhanced chemiluminescence system (Thermo scientific) and Amersham Imager 600 (GE healthcare). Band density was analyzed using this imager.

**Statistics and reproducibility.** Data were presented as the means ± s.e.m. (standard error of the mean) of all experiments with *n* = number of biological replicates. For comparison of samples, data were presented as standard deviation in each group and were evaluated with a two-way ANOVA followed by Tukey's multiple comparisons test or a one-way ANOVA with Dunnett's multiple comparisons test using PRISM 8 GraphPad (version 8.2.0). Statistical significance was obtained by comparing the measures from control group and each treated group. A value of *P < 0.05, **P < 0.001, ***P < 0.0001, ****P < 0.00001 was considered statistically significant.

**Reporting summary.** Further information on research design is available in the Nature Research Reporting Summary linked to this article.

## Data availability

The data-sets generated here were uploaded under the accession GSE161665 (ChIP-seq in GSE161663 and RNA-seq in GSE161664). In addition, we analyzed the ChIP-seq for STATs from GSE31477, CTCF from GSE101051, H3K27ac and H3K4me1 from human lung tissues from GSE143115 and GSE142958, and DNase I hypersensitive (DHS) data from human lung tissues and SAECs from GSE90364 and GSE29692, respectively. RNA-seq data from other types of lung cells, BEAS-2B cells and AT2s, was obtained from GSE148829 and GSE152586, respectively.

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

## Acknowledgements

This work was supported by the Intramural Research Programs (IRPs) of National Institute of Diabetes and Digestive and Kidney Diseases (NIDDK) and National Center for Advancing Translational Sciences (NCATS). We thank Ilhan Akan, Sijung Yun, and Harold Smith from the NIDDK genomics core for NGS and NCATS Chemical Genomics Center Team for the JAK inhibitors. This work utilized the computational resources of the NIH HPC Biowulf cluster (http://hpc.nih.gov).

## Author contributions

H.K.L.: project conception, experimental design and execution, data analysis, preparation of figures, writing manuscript; O.J.: experimental design and execution; L.H.: project conception, experimental design, data analysis, preparation of figures, writing manuscript.

## Competing interests

The authors declare no competing interests.
