## [Peer Review File · Communications Biology]

Reviewers' comments:

Reviewer #1 (Remarks to the Author):

The current manuscript by Lee et al presents interesting hypothesis and data about the activation of interferon-stimulated transcriptomes and ACE2 isoforms in human airway epithelium and its inhibition by Janus Kinase inhibitors. The data is interesting and is presented well. Unfortunately, authors do not go too much in depth leading to the impression of a superficial study that only focuses on observations and not mechanisms.

Major comments:

1. All interferon types (I, II and III) lead to induction of ACE2 expression. These interferons could be secreted by very different cell types and may have very different pathological outcome of the disease. Please discuss this data in the context of disease pathology and possible mechanisms.
2. An underlying assumption in this study is that ACE2 levels correspond to higher virus replication levels and subsequent severe disease. First, mRNA abundance is not always representative of protein levels and second, increased ACE2 levels may not predict a more severe disease (see PMID: 32971472).
3. Also, in lung parenchyma, ACE2 protein is found on the apical surface of a small subset of alveolar type II pneumocytes. Justify your selection of tissues in this regard.
4. While the Chip-seq data on H3K27ac, H3K4me1, and H3K4me3 is informative and is a major focus in this study, there is very little knowledge gained from these experiments in the absence of functional assays that test the importance of these markers in COVID-19 disease progression.
4. The data on JAK inhibitors is interesting but would be bolstered by including and varying the timing of these treatments. Up-regulation of interferon during initial infection may be beneficial for the host whereas its later up-regulation would lead to an increase in the severity of symptoms (e.g. cytokine storm).

Reviewer #2 (Remarks to the Author):

The study conducted by Lee et al. presents great importance for the understanding at molecular level of how key elements involved in immunopathology of SARS-CoV-2 infection may be regulated.

Manuscript is clear and concise, displaying results in well diagrammed figures and comprehensive supplementary tables. However, it is missing some data/analysis or amendments are need. The major concern about the work rely on data exploration of transcriptomes, which is limited. My comments/suggestions throughout the work were listed as follow:

1) Title: replace "epithelium" by "cells"

2) Results:

2.1. Lines 63-64: The scale of y-axis and bars in Fig1a and Fig1c is quite similar for me, it is hard to see difference in the graphs as are currently presented. Therefore, without more explanation in text or legend, the sentence is true only for STAT1, not for ISG15 neither for ISG20. As a transcription factor, I found expression values for STAT1 incredibly too high, please check or provide discussion about this. In Expression Atlas EMBL-EBI webtool, a search for STAT1 expression in different conditions (including infections or cytokine treatment) presented lower levels and smaller amplitudes.

2.2. Lines 86-87: it would be helpful to the readers provides the definitions of chromatin marks (placed in parenthesis) in the figure legend.

2.3. Lines 126-127: full-length of ACE2 doesn't mean all exons? Why only ex1a, ex1b and ex9 are presented?

2.4. Lines 134-141: only 8 lines to describe a transcriptome data. Please go deeper into classical transcriptome analysis and show up the richness of your dataset. RNA-seq data was properly used for ACE2 isoforms part, but it was underexplored as a transcriptome.

3. Methods:

- 3.1. Cell culture: provide how many independent primary cultures were used as biological replicates.
- 3.2. Line 173: replace "were treated" by "were added"
- 3.3. Line 240: exclude "wild-type"; replace "mutant" by treated or treatment.
4. Figure Legends and Figures: text in some legends should be more detailed, please provide information in a way that a non-specialist reader can understand without consulting external material.
 - 4.1. Fig. 1a-c: please use colors for each treatment instead of gray scale in graph bars; improve the y axis scale, it is hard to see difference between plot A and plot C.
 - 4.2. Fig.1d: legend to these plots are insufficient, please explain the plots; contextualize CELF1.
 - 4.3. Fig.1f: y axis – relative activity compare to what? Please provide a correlation analysis between RNA-seq data (fig.1e) and qRT-PCR.
 - 4.4. Fig.2: title legend – replace "epithelium" by "cells".
 - 4.5. Fig2a plot/legend: differentiate chromatin marks from cell types, they are placed in a way that seem to be the same thing.
 - 4.6. Figure 4: in a-b, i) provide a correlation analysis between RNA-seq and qPCR and ii) I would like to know the Ct values for GAPDH for all treatments; in c-d, would be interesting show a negative control gene for all these treatments, i.e., no matter what treatment, the expression level remains the same.
 - 4.7. Fig 4 c-d legend: in line 378, add "and qRT-PCR"; the statistical analysis mentioned in this item refers only to "D" (qRT-PCR), right? For RNA-seq please provide the correct text for statistical analysis (statistical model applied by DESeq2 and FDR value)
 - 4.8. Fig.4e legend: contextualize FAR1
 - 4.9. Figure 5: Overall, the transcriptome analyses will be better represented in classical graphs output from GSEA packages, displaying module activity and over representation analysis of modules and gene networks analysis to show the most connected genes in each scenario. Please perform the proper exploration of RNA-seq data. Many R packages are available for that.
5. Supplementary tables: table 1 – the data compiled on this table is very interesting and meaningful, please provide fold change and FDR values compared to Untreated (data on columns A-I) and IFN-b (data on columns K-N). It would be interesting use all or some of these data to show as figure, a heatmap using statistically relevant genes for example. Table 2: correct the number, it placed as supp table "1" instead of 2.

Reviewers' comments:

Reviewer #1 (Remarks to the Author):

The current manuscript by Lee et al presents interesting hypothesis and data about the activation of interferon-stimulated transcriptomes and ACE2 isoforms in human airway epithelium and its inhibition by Janus Kinase inhibitors. The data is interesting and is presented well. Unfortunately, authors do not go too much in depth leading to the impression of a superficial study that only focuses on observations and not mechanisms.

Response

We thank the reviewer for the positive comments. We have now expanded the study and analyzed the RNA-seq data in great depth. This permitted us to comprehensively identify the genetic programs induced by the different interferon classes and their response to JAK inhibitors (lines 71 to 99, 175 to 183). We also extended this part of the discussion (lines 233 to 259). We also expanded our findings on the dACE isoform and biological implications (lines 202 to 222). The identification of alternative start sites, exons and additional splice forms requires great sequencing depth that is normally achieved with bulk RNA-seq. We highlighted this (lines 221 to 226) as scRNA-seq studies failed to identify the novel dACE2 transcript. We also extended our analysis to (RIG-I)-like receptors, which sense RNA are critical for interferon production in alveolar cells infected by SARS-CoV-2. Our data sets revealed that some of these genes, such as LGP2 (DHX58), are under interferon control and we included these findings (lines 180 to 183).

Major comments:

1. All interferon types (I, II and III) lead to induction of ACE2 expression. These interferons could be secreted by very different cell types and may have very different pathological outcome of the disease. Please discuss this data in the context of disease pathology and possible mechanisms.

Response

Although we investigated all three types of interferons and other cytokines (GH, IL6 and IL7), our work emphasized the impact of type I interferons (α and β) on gene activation in bronchial secretory alveolar cells (lines 233 to 236). We have discussed our work in the context of virally induced interferon production in alveolar cells and the clinically use of JAK inhibitors to suppress molecular consequences of cytokine storms (lines 175 to

183, 250 to 259). We also discussed in greater depth the current status of ACE2 and the novel dACE2 (lines 202 to 222).

2. An underlying assumption in this study is that ACE2 levels correspond to higher virus replication levels and subsequent severe disease. First, mRNA abundance is not always representative of protein levels and second, increased ACE2 levels may not predict a more severe disease (see PMID: 32971472).

Response

Yes, based on several scRNA-seq studies it had been suggested that interferon-induced ACE2 mRNA levels would result in higher protein levels and infection rates and more disease¹⁻⁶. However, based on our study and work by others⁷⁻⁹ it becomes clear that interferons do not significantly induce expression of the full length biologically active ACE2 but a short form (dACE2) that lacks domains needed for SARS-CoV-2 binding. Thus, this induction appears to have no known biological significance.

We carefully read the paper³ referenced by the reviewer and it is clear that the RNA levels do not necessarily match the ACE2 protein levels. However, since scRNA-seq data had been analyzed and the transcripts had not been mapped to individual exons, it is not clear to what extent this study evaluated the full length ACE2 mRNA versus the novel dACE2 transcript that originates from a promoter in intron 9 and that does not yield a protein binding SARS-CoV-2. We included this in the discussion (lines 219 to 222).

We also conducted western blot analyses and identified the full-length ACE2 and the putative dACE2 short form in Supplementary Fig. 4. While there is no evidence that dACE2 has any measurable function, further virology studies might be needed.

3. Also, in lung parenchyma, ACE2 protein is found on the apical surface of a small subset of alveolar type II pneumocytes. Justify your selection of tissues in this regard.

Response

We used small airway epithelial cells (SAECs) that, like human bronchial epithelial cells (HBEs), differentiate in air liquid interphase (ALI) culture systems and form tight junctions and displaying cilia. These are primary cells and therefore most likely reflect the in vivo situation as reliably as possible. We discussed this (lines 233 to 243).

4. While the Chip-seq data on H3K27ac, H3K4me1, and H3K4me3 is informative and is a major focus in this study, there is very little knowledge gained from these experiments

in the absence of functional assays that test the importance of these markers in COVID-19 disease progression.

Response

Yes, the identification of activating histone marks aided us in predicting candidate regulatory elements, including enhancers, for the entirety of genetic programs activated by interferons. To investigate the in vivo significance of these elements would require their homozygous deletion in primary cells through genome editing technologies, a daunting task that could take years. This is clearly beyond the scope of this work. Having said this, our ChIP-seq data sets from primary cells treated with interferons and JAK inhibitors point more directly towards a functional significance as shown in our study.

5. The data on JAK inhibitors is interesting but would be bolstered by including and varying the timing of these treatments. Up-regulation of interferon during initial infection may be beneficial for the host whereas its later up-regulation would lead to an increase in the severity of symptoms (e.g. cytokine storm).

Response

Based on these suggestions, we conducted additional experiments and analyzed ACE2 expression over a time course of 12 hours. These data are shown in Figure 6 (lines 184 to 195). We also analyzed the ability of JAK inhibitors to mitigate ACE2 expression throughout the 12 hour time window.

Reviewer #2 (Remarks to the Author):

The study conducted by Lee et al. presents great importance for the understanding at molecular level of how key elements involved in immunopathology of SARS-CoV-2 infection may be regulated.

Manuscript is clear and concise, displaying results in well diagrammed figures and comprehensive supplementary tables. However, it is missing some data/analysis or amendments are need. The major concern about the work rely on data exploration of transcriptomes, which is limited. My comments/suggestions throughout the work were listed as follow:

Response

We thank the reviewer for the constructive comments. We have now extended our analyses of genetic programs activated in primary bronchial alveolar cells by interferons and other cytokines (lines 74 to 99, 175 to 183). We have also added experiments (new Figure 6) on the kinetics of ACE2 expression (lines 184 to 195).

1) Title: replace “epithelium” by “cells”

Response

We changed “epithelium” to “epithelial cells”.

2) Results:

2.1. Lines 63-64: The scale of y-axis and bars in Fig1a and Fig1c is quite similar for me, it is hard to see difference in the graphs as are currently presented. Therefore, without more explanation in text or legend, the sentence is true only for STAT1, not for ISG15 neither for ISG20. As a transcription factor, I found expression values for STAT1 incredibly too high, please check or provide discussion about this. In Expression Atlas EMBL-EBI webtool, a search for STAT1 expression in different conditions (including infections or cytokine treatment) presented lower levels and smaller amplitudes.

Response

We changed the scale of the y-axis from log values to raw linear values, which highlights the differences. We have also analyzed the RNA-seq data in great depth and show unique responses of individual ISGs (Supplementary Figures 1 and 3) (lines 74 to 99). For example, induction by IFN γ can be more than 2000-fold (IFIT1) and as little as 8-fold (IFITM3). We have also integrated our ChIP-seq data that reveal the enhanced presence of enhancer marks in ISGs upon interferon induction (Supplementary Figure 5).

The values of RNA-seq data can differ depending on the analysis pipeline and final type of values, for example FPKM, RPKM, TPM, or normalized read count from DESeq2 that has been used recently in many papers and also in our study. So, it is not possible to directly compare values from different analyses. We used RNA-seq data from other cell types¹⁰ and STAT1 levels are similar between cell types.

2.2. Lines 86-87: it would be helpful to the readers provides the definitions of chromatin marks (placed in parenthesis) in the figure legend.

Response

We specified the chromatin marks that were investigated (lines 123 to 124). We used H3K27ac (activate loci), H3K4me1 (enhancers) and H3K4me3 (promoters). This information has also been added in the figure legend.

2.3. Lines 126-127: full-length of ACE2 doesn't mean all exons? Why only ex1a, ex1b and ex9 are presented?

Response

The structure of the entire ACE2 locus, including all exons and introns, is shown in Fig. 2a and the top panel of Fig. 2b. The lower panel of Fig. 2b focuses on the promoter sites and exons relevant to this study. Exons 1a and 1b (ex1a and ex1b) are part of the full length ACE2 transcript encoding the native ACE2 (first ATG is located in ex1b). The first exon (ex1c) of the new isoform is located between exons 9 and 10 (ex9 and ex10) and we adopted the previously described nomenclature⁹. The two TaqMan probe (ex9 and ex1c) distinguish the expression of the full-length ACE2 and the new isoform dACE2.

2.4. Lines 134-141: only 8 lines to describe a transcriptome data. Please go deeper into classical transcriptome analysis and show up the richness of your dataset. RNA-seq data was properly used for ACE2 isoforms part, but it was underexplored as a transcriptome.

Response

We thank the reviewer for this suggestion. As part of our study we had generated RNA-seq and ChIP-seq data from primary bronchial alveolar cells stimulated with four interferons α , β , γ and λ , GH, IL6 and IL7. We also conducted RNA-seq and ChIP-seq studies to determine the efficacy and mechanism of JAK inhibitors. We have now analyzed these data sets in greater depth (lines 74 to 99, 154 to 159, 175 to 183). In doing so, we further identified and characterized genetic programs activated in bronchial cells by interferons. We also demonstrated the efficacy of JAK inhibitors to suppress interferon induction, both on the levels of gene expression and activating histone marks.

3. Methods:

3.1. Cell culture: provide how many independent primary cultures were used as biological replicates.

Response

The number of biological replicates is provided in the figure legend for each graph.

3.2. Line 173: replace “were treated” by “were added”

Response

We replaced the word.

3.3. Line 240: exclude “wild-type”; replace “mutant” by treated or treatment.

Response

We corrected the word.

4. Figure Legends and Figures: text in some legends should be more detailed, please provide information in a way that a non-specialist reader can understand without consulting external material.

4.1. Fig. 1a-c: please use colors for each treatment instead of gray scale in graph bars; improve the y axis scale, it is hard to see difference between plot A and plot C.

Response

We have now used different colors for each treatment and the scale of y-axis changed from \log_{10} values to linear values.

4.2. Fig.1d: legend to these plots are insufficient, please explain the plots; contextualize CELF1.

Response

We added more explanation for the plots in figure legend.

4.3. Fig.1f: y axis – relative activity compare to what? Please provide a correlation analysis between RNA-seq data (fig.1e) and qRT-PCR.

Response

Figure 1e shows the relative read numbers of RNA-seq for each exon and Figure 1f confirmed the finding via qRT-PCR for full-length ACE2 and the novel dACE2.

Therefore, in Figure 1f, relative fold activity of cytokine treated groups was compared to the control. We added this in figure legend.

4.4. Fig.2: title legend – replace “epithelium” by “cells”.

Response

We replaced “epithelium” to “epithelial cells”.

4.5. Fig2a plot/legend: differentiate chromatin marks from cell types, they are placed in a way that seem to be the same thing.

Response

Thank you for pointing this out. We edited and clarified the figure. Cell types and chromatin marks are now in separate ‘columns’.

4.6. Figure 4: in a-b, i) provide a correlation analysis between RNA-seq and qPCR and ii) I would like to know the Ct values for GAPDH for all treatments; in c-d, would be interesting show a negative control gene for all these treatments, i.e., no matter what treatment, the expression level remains the same.

Response

i) We clarified the results from RNA-seq and qRT-PCR in the figure legend.

ii) We obtained Ct values of 20-21 for human GAPDH with TaqMan probe (Hs02786624_g1, Thermo Fisher scientific) and there was no difference by treating with cytokines or JAK inhibitors.

We conducted qRT-PCR to identify the expression levels of full-length ACE2 and dACE2. We had already determined the ACE2 expression levels, stimulated by cytokines and inhibited by JAK inhibitors, using RNA-seq. When conducting qRT-PCR, we determined the ACE2 and STAT1 levels as positive controls and a mixture without template as a negative control (data not shown).

4.7. Fig 4 c-d legend: in line 378, add “and qRT-PCR”; the statistical analysis mentioned in this item refers only to “D” (qRT-PCR), right? For RNA-seq please provide the correct text for statistical analysis (statistical model applied by DESeq2 and FDR value)

Response

Figure 4c-d are graphs from RNA-seq data, but not from qRT-PCR experiments. FDR values for between control and experimental (cytokine/Jak inhibitor-treated) groups are in Supplementary Table 3, 9 and 10. Statistical significance for selected values from multiple RNA-seq data were calculated using PRISM 8 GraphPad.

4.8. Fig.4e legend: contextualize FAR1

Response

The FAR1 locus was used as a ChIP-seq control and we added this in the figure legend.

4.9. Figure 5: Overall, the transcriptome analyses will be better represented in classical graphs output from GSEA packages, displaying module activity and over representation analysis of modules and gene networks analysis to show the most connected genes in each scenario. Please perform the proper exploration of RNA-seq data. Many R packages are available for that.

Response

The main purpose of the paper was the identification and analysis of the novel dACE2 and its regulation as well as a global analysis of genes induced by cytokines and the therapeutic effect of JAK inhibitors. At this point we don't think that network analyses would enhance the message of the paper. Since all data sets have been deposited in public repositories, other researchers have the opportunity to mine the data according to their needs.

5. Supplementary tables: table 1 – the data compiled on this table is very interesting and meaningful, please provide fold change and FDR values compared to Untreated (data on columns A-I) and IFN-b (data on columns K-N). It would be interesting use all or some of these data to show as figure, a heatmap using statistically relevant genes for example. Table 2: correct the number, it placed as supp table “1” instead of 2.

Response

The Supplementary Table 1 is a summary for specific gene set from Supplementary Tables 2-10. The fold change and FDR values between untreated and each experimental group are in Supplementary Tables 2-10, respectively.

Of the data in Supplementary Table 1, ACE2 and STAT1 are presented as a graph in Figure 1.

1. Chua, R.L. *et al.* COVID-19 severity correlates with airway epithelium-immune cell interactions identified by single-cell analysis. *Nat Biotechnol* **38**, 970-979 (2020).
2. Katsura, H. *et al.* Human Lung Stem Cell-Based Alveolospheres Provide Insights into SARS-CoV-2-Mediated Interferon Responses and Pneumocyte Dysfunction. *Cell Stem Cell* **27**, 890-904.e8 (2020).
3. Ortiz, M.E. *et al.* Heterogeneous expression of the SARS-Coronavirus-2 receptor ACE2 in the human respiratory tract. *EBioMedicine* **60**, 102976 (2020).
4. Sajuthi, S.P. *et al.* Type 2 and interferon inflammation regulate SARS-CoV-2 entry factor expression in the airway epithelium. *Nat Commun* **11**, 5139 (2020).
5. Youk, J. *et al.* Three-Dimensional Human Alveolar Stem Cell Culture Models Reveal Infection Response to SARS-CoV-2. *Cell Stem Cell* **27**, 905-919 e10 (2020).
6. Ziegler, C.G.K. *et al.* SARS-CoV-2 Receptor ACE2 Is an Interferon-Stimulated Gene in Human Airway Epithelial Cells and Is Detected in Specific Cell Subsets across Tissues. *Cell* (2020).
7. Blume, C. *et al.* A novel ACE2 isoform is expressed in human respiratory epithelia and is upregulated in response to interferons and RNA respiratory virus infection. *Nat Genet* **53**, 205-214 (2021).
8. Ng, K.W. *et al.* Tissue-specific and interferon-inducible expression of nonfunctional ACE2 through endogenous retroelement co-option. *Nat Genet* (2020).
9. Onabajo, O.O. *et al.* Interferons and viruses induce a novel truncated ACE2 isoform and not the full-length SARS-CoV-2 receptor. *Nat Genet* (2020).
10. Lee, H.K. *et al.* Immune transcriptomes of highly exposed SARS-CoV-2 asymptomatic seropositive versus seronegative individuals from the Ischgl community. *Sci Rep* **11**, 4243 (2021).

REVIEWERS' COMMENTS:

Reviewer #1 (Remarks to the Author):

Authors have included additional data and discussion to support their conclusions. They have satisfactorily responded to most of the reviewers' comments. Only one minor point needs to be explained better.

dACE2 encodes a non-functional protein, which does not bind SARS-CoV-2. How are the levels of dACE2 transcript relevant to SARS-CoV-2 infection? What are the consequences of an interplay of interferons, ACE-2 and dACE-2 that is relevant to the infection and pathology of SARS-CoV-2?

Reviewer #2 (Remarks to the Author):

The revised version of manuscript was modified and improved accordingly. All my comments and questions were properly addressed by the authors in rebuttal letter. I have no further comments.